# Diversity of Adult Neural Stem and Progenitor Cells in Physiology and Disease

**DOI:** 10.3390/cells10082045

**Published:** 2021-08-10

**Authors:** Zachary Finkel, Fatima Esteban, Brianna Rodriguez, Tianyue Fu, Xin Ai, Li Cai

**Affiliations:** Department of Biomedical Engineering, Rutgers University, Piscataway, NJ 08854, USA; zf99@rutgers.edu (Z.F.); fce10@scarletmail.rutgers.edu (F.E.); br444@scarletmail.rutgers.edu (B.R.); tf274@scarletmail.rutgers.edu (T.F.); xa16@rutgers.edu (X.A.)

**Keywords:** central nervous system (CNS), ependymal cells, neural stem and progenitor cells (NSPC), NG2+ cells, neurodegenerative diseases, regenerative medicine, retina injury, spinal cord injury (SCI), traumatic brain injury (TBI)

## Abstract

Adult neural stem and progenitor cells (NSPCs) contribute to learning, memory, maintenance of homeostasis, energy metabolism and many other essential processes. They are highly heterogeneous populations that require input from a regionally distinct microenvironment including a mix of neurons, oligodendrocytes, astrocytes, ependymal cells, NG2+ glia, vasculature, cerebrospinal fluid (CSF), and others. The diversity of NSPCs is present in all three major parts of the CNS, i.e., the brain, spinal cord, and retina. Intrinsic and extrinsic signals, e.g., neurotrophic and growth factors, master transcription factors, and mechanical properties of the extracellular matrix (ECM), collectively regulate activities and characteristics of NSPCs: quiescence/survival, proliferation, migration, differentiation, and integration. This review discusses the heterogeneous NSPC populations in the normal physiology and highlights their potentials and roles in injured/diseased states for regenerative medicine.

## 1. Introduction

During development, neural stem cells (NSCs) are responsible for the formation of the central nervous system (CNS). Initially, NSCs, also called neuroepithelial cells, differentiate into radial glial cells and proliferate into pools of neural progenitor cells (NPCs) [1]. NSC refers to an uncommitted cell with differentiation potential into the neurons and glia of the CNS. NSC is defined by two essential characteristics: self-renewal and multipotency [2]. These neural stem and progenitor cells (NSPCs) represent both populations and are established as the only self-renewing cell type in the adult CNS. NSPCs migrate and differentiate into highly specified networks of neurons via neurogenesis, and oligodendrocytes and astrocytes are generated via gliogenesis [3,4] (Figure 1). Thus, NSPCs are a major research thrust in the field of regenerative medicine. Extrinsic and intrinsic factors such as neurotrophic/growth factors, transcription factors, and canonical pathways guide neurogenesis and gliogenesis during development and adulthood.

For the past 50 years, the topic of endogenous adult neurogenesis has been highly debated. This began with the initial discovery of adult mammalian neurogenesis in 1962 by Joseph Altman and has continued with noteworthy publications supporting the existence or non-existence of adult neurogenesis in mammals [5]. In the adult CNS, neurogenesis plays a primary role in essential processes such as learning, memory, maintenance of tissue homeostasis, and many others. 

Heterogeneous populations of NSPCs exist in the neurogenic niches of the brain, spinal cord, and retina. Primary NSPCs are found in the subventricular zone (SVZ) and subgranular zone (SGZ) of the brain and include radial glial-like cells, NG2+/oligodendrocyte progenitor cells (OPCs), and Foxj1+ ependymal cells. Both OPCs and ependymal cell populations can be found in the spinal cord. In the adult retina, potential sources of NSPCs include Müller glia cells and the ciliary epithelium (CE).

NSPC response to CNS injury is extraordinarily complex and dependent upon the extent and location of injury. Injuries are most often contusion or blunt force-based and primarily result from sporting or vehicular accidents. Traumatic brain injuries (TBI) habitually damage two central niches: SGZ of the hippocampus and SVZ of the lateral ventricles. Damage to these regions can result in consequences including aberrant migration of NSPC progeny cells, incorrect dendritic branching, enhanced progenitor cell proliferation, ineffective integration of cells into networks of tissue, and many others. Spinal cord injury (SCI) may affect the neurogenic niche of the central canal resulting in differing contributions of NSPC populations to the glial scar. In addition, large differences in injury pathophysiology occur as a direct result of injury-mediated proliferation and altered differentiation.

In the eye, retinal injury results from chemical or mechanical damage and is highly dependent on NSPC activity. Traumatic mechanical injury of the eye results in severe morphological and functional changes in the eye structure including retinal detachment in humans [7]. Common retinal degenerative diseases include retinitis pigmentosa (RP), age-related macular degeneration (AMD) and glaucoma. Retinal degeneration affects photoreceptors, retinal ganglion cells and retinal pigment epithelium (RPE) to cause vision loss at varying degrees and eventual blindness. 

Adult neurogenesis in heterogeneous NSPC populations has been implicated in demyelinating, inflammatory, and neurodegenerative conditions such as Alzheimer’s disease (AD), Parkinson’s disease (PD), multiple sclerosis (MS), and schizophrenia [8,9,10]. Early onset AD has been largely attributed to two genetic mutations, APP and presenilins (PS) [11]. Gene knock-out and knock-in mouse models show decreased neurogenesis, learning, and memory associated with upregulation of PS genes [12]. PD is characterized by progressive degeneration of dopaminergic neurons and PD-associated transgenic animal models have shown increased neurogenesis in dopaminergic neurons [13]. MS is defined by oligodendrocyte loss and axonal degeneration/demyelination [14]. A reduction in progenitor proliferation in the SVZ was observed in the lesion model of MS. Neuronal loss or axonal damage is characteristic of these conditions, thus modulation of adult neurogenesis, the generation of new neurons, has been proposed as a prospective treatment.

Neurogenic activity of the brain, spinal cord, and retina may facilitate the generation of functional networks of integrated tissue in damaged or diseased areas. Overall, NSPCs play an essential role in injuries or degenerative disorders that are largely affected by neurogenesis and disruptions in cell behavior such as traumatic brain injury (TBI), spinal cord injury (SCI), retinal injury, multiple sclerosis (MS), and schizophrenia [8,9,10]. Thus, an in-depth understanding of neurogenesis throughout the CNS will facilitate effective stem cell oriented therapeutic development.

## 2. Neural Stem/Progenitor Cells

The adult NSPCs (e.g., progenitor cells, neuroblasts, ependymal cells, NG2+ glia) are present in the stem cell niches of the brain, spinal cord, and retina. Major cell types present in the general NSPC niche include neurons, oligodendrocytes, astrocytes, pericytes, and endothelial cells. Neural stem cells primarily reside in the neural niches of the CNS, whereas progenitor cells can be found throughout the CNS due to increased migratory capacity [15,16,17,18].

Additional contributors to the microenvironment of NSPCs in CNS niches include cerebrospinal fluid (CSF), the extracellular matrix (ECM), and vasculature. The CSF consists of neurotrophic/growth factors, transcription factors, and ECM molecules required for NSPC guidance and is important for cell migration, morphogenesis, growth, and development [19]. The ECM provides mechanical support and regulates extracellular signaling environments. Moreover, proteoglycan and glycoprotein composition varies to influence signaling and bioavailability, motivating NSPC behavior within the stem cell niche [20]. 

Cellular cross talk between the stem cells and specified cell types contribute to the symphony of cascading signals regulating NSPC behavior. NSPC populations in the stem cell niche are highly regulated to produce neuronal or glial lineage cell types [21]. The vasculature also regulates neurogenesis in the adult CNS by transport of infiltrating biochemical signals to interact with NSPCs [22]. In this way, intrinsic and extrinsic signals regulate neurogenesis, generated via cross talk with cells, vasculature, ECM via external forces, and CSF in the neural niche. Intrinsic signals include master transcription factors such as Sox2 and REST [23]. Extrinsic signals include neurotrophic/trophic and growth factors, neurotransmitters, and signaling pathways such as Wnt and Notch.

When networks of neural cell types responsible for a regulated signaling microenvironment are damaged, NSPCs exhibit extreme behavior [24]. This is due to distinctly different signals or lack of signals required to regulate pools of active or quiescent NSPCs. Traumatic injury stimulates NSPCs to proliferate rapidly and produce cells which contribute to the glial scar border and upregulate angiogenesis in addition to neurogenic activities [10]. Preferential survival of transplanted NSCs was observed in geographical areas of high-density vasculature, which is said to play an essential role in the survival and maintenance of NSPCs in the injured spinal cord [22].

NSPCs often generate new non-functional networks of cells in response to injury which inhibits neural regeneration [25]. Altered niche activity may contribute to segregation of the injury but does not lead to regeneration of functional tissue. Differences in traumatic injury type and grade in the CNS result in significant changes in neurogenesis in one or more niches [24]. The heterogeneity of cell populations affected by traumatic injury result in clinical inconsistencies between cases. Further, the neurogenic niches of the brain, spinal cord, or retina exhibit regionally distinct niche composition before and after traumatic injury (Figure 2).

### 2.1. Adult NSPCs in the Brain

The mammalian brain contains two primary neurogenic niches, i.e., the SGZ of the hippocampus and the SVZ of the lateral ventricles [31]. The hypothalamus serves as a third neurogenic niche conserved in some species but is nonexistent in humans [32]. Each distinct niche contains specific populations of NSPCs and differing functions. 

The hippocampal neurogenic niche is present at the base of the hippocampus within the dentate gyrus (DG) in the SGZ (Figure 2A). In this niche, NSPCs are required for maintenance of the hippocampal tissue homeostasis, learning, and memory. Major stem cells in this neurogenic niche are radial glial-like cells (RGLs) which maintain neurogenic activity into adulthood [33]. Key cell types include OPCs, neuroblasts, immature/mature neurons, and oligodendrocytes. As a note, OPC populations in this niche include NG2+ cells. 

The neurogenic niche along the walls of the lateral ventricles is located in the SVZ (Figure 2A). The lateral ventricle niche can be separated into two different geographical regions in the tissue: 1. dorsal, 2. lateral. Both dorsal and lateral components are in direct contact with pools of CSF, where the ependymal cell layer serves as a border between CSF and niche NSPCs [34]. This allows regulated contact between the ventricular cavities and undifferentiated progeny. Internal mechanisms direct NSPC behavior via fluid flow of CSF in the lateral ventricles [19]. NSPCs include astrocytes, neuronal/and glial progenitor subtypes, and neuroblasts [35,36,37]. Progenitors can be subdivided into further populations based on gene mapping analysis in both domains. Transcriptional patterning in temporal and spatial arrangements shows distinct NSPC populations [38]. Differential gene expression is driven by cell niche based signaling. Major signals include the Wnt/B-catenin and sonic hedgehog (Shh) pathway and are important to maintain regulatory behavior [39]. 

Adult neurogenesis in the hippocampus is dictated by intrinsic and extrinsic cues [40]. Signaling is initiated by surrounding cell types and vasculature in addition to master transcription factors Oct4, Sox2, and CREB. Signals from the Notch, Wnt, Shh, and other pathways direct neurogenesis in the SGZ. 

The rostral migratory stream is a migration pathway for neuroblasts from the SVZ to the olfactory bulb and is present in some mammalian species. Conserved signaling pathways direct differentiation and integration of specified neurons and glia into the olfactory bulb. However, this is present to a lesser extent in larger mammalian species such as humans. Migrating neuroblasts from the hippocampal niche have been documented in rodent models to contribute to olfactory bulb mature cell types [41]. However, in human and primate models these cells are instead generated in the striatum. Damage to the neural niche of the hippocampus has been associated with cognitive deficits in learning and memory.

The hypothalamus neurogenic niche is located near the lateral ventricles below the SVZ, also called the periventricular zone [42]. Major cell types in this niche include hypothalamic ribbon cells lining the outer wall and monocytes which may present neurogenic potential. Three populations of NSCs have been found in the hypothalamus of animal models including mouse, rat, and monkey including tanycytes, ependymal cells, and small stellate cells [15]. These populations generate neurons and glia throughout life in the hypothalamic parenchyma. Neurogenesis in this region occurs at a lesser incidence in comparison with the two classic niches, hippocampal SGZ and lateral ventricles SVZ. This may translate into functional significance in murine models via control of energy metabolism.

In the injured brain, specific regulation of quiescence/survival in NSPCs has been attributed to the small glycoprotein lactadherin, growth factors vascular endothelial growth factor (VEGF), fibroblast growth factor-2 (FGF2), and Notch and Wnt pathways [39,43,44]. Proliferation is regulated by lactadherin, amyloid precursor protein, neurotrophic factor Tumor necrosis factor alpha (TNFa), growth factors FGF2, and VEGF, chemokine CX3CL1, and pathways Shh, Notch, and Wnt [45]. Migration is regulated by growth factor VEGF, chemokines CCR2 and CX3CL1, and the Wnt pathway [44]. Differentiation is regulated by growth factors FGF2 and VEGF, chemokines CCR2 and CX3CL1, as well as Notch and Shh pathways. Integration is regulated by growth factor VEGF and chemokine CX3CL1 [43]. Injury-induced or altered signals contribute to the enhanced proliferation, aberrant progenitor migration, ineffective integration, and reduced dendritic branching observed in TBI and SCI. 

Using a combination of transgenic mouse model and single-cell RNA-seq analysis, distinct adult NSPC populations were identified in the SVZ [46]. In this study, GFP+ cells represent Nestin+ stem cell populations in the adult. Four groups of NSCs and three groups of progenitor cells were characterized with in vivo and in vitro RNA-seq studies of the SVZ neurogenic niche [46]. Immunostaining and imaging analysis revealed distinct subgroups of cells separated by signal intensity: high GFP, low GFP and no GFP, and co-labeled with specific markers such as DCX and GLAST. Further, RNA-seq analysis isolated cells into profiles of quiescent and active stem cells in addition to stem cell markers, e.g., Sox2, Ascl1, and DCX. Groups of cells are also separated anatomically, further supporting the existence of distinct populations. NSPC heterogeneity has also been demonstrated using stem cell markers including Gli1 and Ascl1 in both dividing and nondividing NSPCs [47]. The utility of these NSPC populations is unknown, but clear differences exist in gene expression profile.

### 2.2. Adult NPSCs in the Spinal Cord

The mammalian spinal cord contains one neurogenic niche in the ependyma of the central canal in which stem cells are present in an undifferentiated and self-renewable state (Figure 2B). The central canal serves as a continuation of the lateral ventricles into the spinal cord, while the ependymal cells serve as the bridge and a major regulatory element between the CSF and the stem cell niche [48]. The central canal neurogenic niche is lined with multiple populations of ependymal cells and CSF contacting neurons [49]. Ependymal cell populations can be further characterized into cells with short basal processes and cells with long extended processes. Other major components of the niche include NG2+ cells, vasculature, astroglial cells, and oligodendrocytes. Populations of progenitors in the spinal cord are indicated by markers Olig2, PDGFRa, and NG2 [50]. In addition, the ependymal cell layer is surrounded by supporting mature cell types, while the layer itself contains astroglial cells, NG2+ cells, and Nestin+ undifferentiated stem cells [49]. In normal physiology, NSPC proliferation is observed in this stem cell niche, indicated by Ki67 antibody staining in numerous studies [51,52]. 

Extrinsic signals guiding adult neurogenesis in the spinal cord include connexin, Notch and Wnt signaling pathways [18,53]. Intrinsic signals include neural progenitor transcription factors Nkx6.1, Pax6, and Olig6 [54,55,56]. These signals cohesively create an environment to control NSPC activity and maintain normal pools of immature and mature cell types in quiescent or active states. During injury or disease, NSPCs are subject to altered specific niche-based signals and exhibit skewed behavior. Thus, the neural niche in the central canal of the spinal cord is incredibly unique and maintained by a delicate balance of intrinsic and extrinsic signals.

SCI affects the NSPC stem cell niche in models of contusive, surgical stab, and slice injury at any anatomical level of the spinal cord [53]. Common clinical SCI disturbs the niche due to equidistant dorsal and ventral positioning of the central canal [24]. NSPCs proliferate after injury and interact with inflammatory signals to produce the glial scar border, a chemical/physical barrier which segregates the injury and prevents additional damage [57]. However, this scar also prevents axonal outgrowth into the site of injury and generation of new cell types within the neural lesion. NSPCs proliferate and differentiate into reactive astrocytes in the injured spinal cord and contribute to the glial scar border. In addition to newly generated progeny, resident astrocytes transition to reactive gliosis state and are recruited to the site of injury, lengthen their processes, and fatten to become the scar border [58]. A multitude of NSPCs in the spinal cord produce progeny of differing lineages to contribute to the glial scar after SCI and TBI. 

Two major cell types have been controversially implicated in the NSPC response to injury and pose high therapeutic potential: NG2+ and ependymal cells. Many published studies are in support of the stem-like character or non-stem-like character of these cells. Both NG2+ cells and ependymal cells have been reported to contribute to the formation of the scar border. More recently, NG2+ cells have been shown to contribute to the generation of neurons in the injured spinal cord [57,59]. We will discuss the heterogeneity of NSPCs after injury with a focus on the activity of NG2+ and ependymal cells in Section 4.

### 2.3. Adult Retinal Stem Cells

Cells from regions of the adult retina such as the retinal pigment epithelium (RPE) [60,61], CE [62,63,64,65,66], Müller glia cells [64,67,68,69], iris pigment epithelium [70,71] and optic nerve [61] show stem cell characteristics to varying degrees in humans and rodents (Figure 2C). Among them, the CE and Müller glia are identified as two main retinal stem cell sources. 

A subpopulation of adult human RPE cells is capable of being activated to become RPE retinal stem cells in vitro and differentiated into multipotent stable RPE or mesenchymal lineages [60]. The optic nerve lamina region (ONLR) in both humans and mice contains a retinal NPC niche [61]. Adult NPCs in the ONLR exhibit multipotency and generate two types of glia: astrocytes and oligodendrocytes. These populations contribute to enable glial replacement and remyelination in adulthood [61]. The derived adult rat iris pigment epithelium (IPE) cells have NSPC properties and can differentiate into rod photoreceptor cells under CRX expression [71]. NeuroD induces human iris cells into rod photoreceptor cells. Moreover, Yuko et al. observed the combination of CRX, RX and NeuroD induces the generation of photoreceptor cells from the derived human IPE cells [70].

Non-pigmented CE cells show stem cell markers and actively proliferate after photoreceptor cell degeneration or retinal ganglion cell injury in the mouse model [62,72]. In the human CE, non-pigmented CE cells are labeled with stem cell markers, e.g., Sox2, Chx10 and Notch1. Non-pigmented CE cells showed proliferative ability under epidermal growth factor (EGF) induction using explants of the human retina [63]. CE cells including the pigmented cells and non-pigmented cells from human and mouse express NSPC cell markers and characteristics in vitro [64]. CE cells can be induced into photoreceptor cells, bipolar cells, retinal ganglion cells and Müller glia cells in the mouse model [65]. In addition, human CE cells can be induced into many types of retinal cells in vitro [66]. 

Müller glial cells are also considered as a primary source of retinal stem cells. Bhatia et al. concluded that retinal Müller glia may perform similar functions ascribed to astrocytes, ependymal cells and oligodendrocytes in other regions of the CNS [64]. Das et al. also stated that Müller glia are the NSCs of the adult retina [67]. They demonstrated that rat Müller glia have potential to generate retinal neurons in vitro and in vivo. Moreover, they proved the role of Notch and Wnt pathways in regulating this activity. Similarly, in mouse models, Müller glia can be reprogrammed into photoreceptors and retinal ganglion cells under certain culture conditions [68]. In adult human eyes, no evidence has been found to suggest that Müller glia possess the retinal neuronal regeneration ability in vivo. However, in vitro, these progenitor-type glia can be induced to proliferate and differentiate into retinal neurons and RPE cells [69]. Human Müller glia-derived stem cells can be differentiated toward the fate of retinal ganglion cell (RGC) precursors using FGF-2 and Notch inhibition [69]. In summary, Müller glia-derived stem cells can function as NSCs and serve as a potential target of therapy for retinal degenerative disease. 

Common retinal diseases/injuries such as retinitis pigmentosa (RP) and age-related macular degeneration (AMD) cause the photoreceptor cell loss and damaged RPE. However, no enhanced differentiation or proliferation was observed after injury [65]. Damaged cells release growth factors and cytokines which cause the Müller glia cell to differentiate, proliferate and express progenitor cell markers [73]. The ability of these proliferating Müller cells to regenerate new neurons and repair the injured retina appears to be extremely limited. Regardless, the multipotent stem cells may generate more functional photoreceptor cells and help with the recovery of vision loss in the RP and AMD via transplantation method [74].

### 2.4. Heterogeneity between CNS Niches

The perivascular stem cell niche is not technically a NSPC niche, but it interacts with cell types and influences NSPC behavior in all niches, thus contributing to the diversity of NSPC behavior observed in the mammalian CNS. In particular, the retina contains sources of NSPCs such as Müller glia and CE. Major factors unique to the retinal niche include CRX, RX and NeuroD. Interestingly, the retina does not contain ependymal cells, a major controversial stem type cell in the brain and spinal cord. However, NG2+ cells can be found in the retina [75]. The brain contains NSPC populations such as radial glial-like cells, OPCs, and ependymal cells. However, these populations and their characteristics vary throughout distinct NSPC niches. Major signals unique to the SGZ and SVZ include Shh pathway and transcription factors CREB and Oct4 [76]. The spinal cord stem cell niche contains both ependymal cells and NG2+ cells. Signals unique to the spinal cord include connexin signaling. The activity and consistency of NG2+ populations vary significantly between the niches of the brain, spinal cord, and retina. Specifically, NG2+ cells in the brain and spinal cord generate oligodendrocyte cell types and consist of glia and pericytes [77]. However, NG2+ cells in the retina consist of microglia and pericytes [75]. Ependymal cells also exhibit a variety of diverse behaviors in neurogenic niches of the brain and spinal cord. These controversial stem-like cells will be discussed in the following sections. 

Understanding the heterogeneity of these stem cell populations and neurogenic niches is necessary to effectively design therapeutics for SCI, TBI, mechanical/chemical injury, and diseased states such as Glaucoma, Retinitis Pigmentosa, demyelinating diseases, and inflammatory conditions.

## 3. Notch1CR2-GFP+ NSPCs in Development and Injury

The canonical Notch signaling pathway is required to regulate the quiescence, proliferation, and differentiation of NSPCs in the CNS [56,78,79,80]. The Cai lab identified a 399-bp cis-element in the second intron of the Notch1 locus (CR2) [81]. In the Notch1CR2-GFP transgenic mouse, CR2 directs the reporter GFP expression in the interneuron progenitor cells. The activities of Notch pathway and NSPCs can be traced by the reporter GFP expression (Figure 3A). The cell fate of GFP tagged interneuron progenitors have been characterized in both normal development and neurological disease/injury conditions, which facilitate the study of the potentials of NSPCs in regenerative medicine [79,80,81,82]. In these studies, the Cai lab has demonstrated that GFP+ NSPCs preferentially differentiate into interneurons of the brain and spinal cord during embryonic development and in adulthood [56,80]. Injury increased the number of GFP+ NSPCs and interneurons at the injury site in a closed head injury model [80]. These results demonstrate that the endogenous NSPCs in the brain proliferate after injury and differentiate into specific cell fates (Figure 3B).

In a more recent study, virus-mediated Gsx1 expression in NSPCs displayed an increased rate of cell proliferation with increased number of GFP+ NSPCs. Gsx1 further promoted neuronal differentiation over glial lineage in the injured spinal cord (Figure 3C). This resulted in an increased number of neurons, reduced reactive astrocytes and glial scar formation, and improved functional recovery [79]. Genetic manipulation of NSPCs is a primary therapeutic approach in the field of regenerative medicine [83]. Many conditions are defined by major cell loss and accompanied by decreased neurogenesis, e.g., SCI, TBI, MS, PD. Engineering NSPCs to increase proliferation and differentiation presents a viable option to promote effective regeneration of lost tissue in the CNS [84]. Gene/cell therapy can be used to express target genes in host cells, e.g., neurons, astrocytes, NSPCs, and oligodendrocytes [85,86]. NSPC specificity can be accomplished via choice of promotor, enhancer, and viral serotype. Common promoters target NSPCs including Nestin, Notch1, NG2, and Sox2. In recent years, forced expression of neurogenic genes (e.g., Ascl1, Gsx1, and Sox11) in stem cell populations promotes cell/tissue regeneration [79,87,88]. Many NSPC subpopulations have been identified, but functional and mechanistic understanding is limited [89]. Transgenic animal models such as the Notch1CR2-GFP allow in vivo investigation of specific NSPC populations and are vital to develop effective therapeutics in the future [56,79,80,81,82].

The Notch1CR2-GFP transgenic animal model serves as a valuable tool to study endogenous NSPCs following traumatic CNS injury [56,79,80,81,82]. Further, NSPCs represent an important cell source for neural regeneration in the adult mammalian CNS [56,79,80,81,82]. For this reason, diversity of NSPC populations (e.g., Nestin+, Notch1+, NG2+, Foxj1+ cells) have become an intensely focused area of research in regenerative medicine for CNS diseases and injuries. Several controversial issues arise and are discussed in the next section.

## 4. Controversial NSPC Populations in Injury/Disease

Adult NSPC populations are composed of diverse cell types in the mammalian CNS, which contribute to growth and regeneration after injury and disease. Several cell types, e.g., ependymal and NG2+ cells have been controversially proposed as stem cells. The stem cell behavior of these populations has been implicated in the injury response and neurodegenerative/demyelinating disorders [90]. These populations are primarily glial producing cells; however, extrinsic and intrinsic factors have been used to modulate the glial fate to neuronal fate for functional recovery after traumatic injury [59,80]. Both populations have also been polemically associated with the glial scar formation after TBI and SCI in mammals [91,92]. In the following sections, we will explore the stemness of these populations in normal physiology and the utility of these populations as a treatment for CNS injury/disease. To appropriately assess ependymal and NG2+ cells as a stem cell, we will use the basic criteria: self-renewal and multipotency [2,93] (Figure 1).

### 4.1. Ependymal Cells

Ependymal cells are neuroglia which line the central canal of the spinal cord and the lateral ventricles of the brain. The epithelial layer (ependyma) acts as a barrier between the CSF and stem cell niche and regulates CSF balance and NSPC activity in central CNS niches [49]. In this layer, ependymal cells project differing length cilia into the CSF and aid in motility, production, and absorption of the CSF [94]. Ependymal cells have been controversially proposed as stem cells in the brain and spinal cord. Contradicting results have been reported regarding the appropriate contribution of ependymal cells to the pathophysiology of SCI and TBI [91,95,96]. This discrepancy is attributed to major differences in CNS injury models, animal models, and quantification techniques [95,97]. Differences in injury models range from damage to the ependymal layer, gray, and white matter to no damage to the ependymal layer but exposure to injury mediators such as glutamate. Further, age of animals used in research may also account for discrepancies in reported ependymal cell ability, as younger animals maintain populations with increased neurogenic activity in comparison with older animals [95].

During development, the ependymal and neuronal cell fates are decided by numerous transcription factors, e.g., Ascl1, Sox10. Non-differentiated pools of progenitors generated by NSCs produce ependymal and neural cell types in a finely tuned spatiotemporal manner [98]. By embryonic day 15.5 (E15.5), the ependymal cell populations can be fully distinguished [99]. In the adult, ependymal cells express Foxj1 and proliferate actively to produce multipotent glial fated cell types such as astrocytes and oligodendrocytes [91]. This does not occur at a high rate or contribute to tumorigenesis in the spinal cord or brain [51]. Pools of ependymal cells also have displayed self-renewal capability, but this capability decreases with age of animal [95]. 

Extrinsic factors guide ependymal cell activity in the adult including neurotrophic factors and Notch and Wnt signaling pathways. Intrinsic factors include transcription factor Foxj1 and nuclear factor IX (NFIX) [94]. Quiescence/survival is regulated by DNA-binding protein inhibitor (Id3) and HES family transcription factor 5 (Hes5) [97]. Proliferation is regulated by Wnt signaling and growth factors [100]. Differentiation is directed by the Geminin superfamily, an antagonist of DNA replication, and NFIX [52,101]. Migration is regulated by NFIX and non-muscle myosin II [48]. Ependymal cells also secrete factors to produce chemical gradients promoting migration of neuroblasts in the SVZ of the lateral ventricles [102] (Table 1). Ependymal cell activity is consistent with the consensus that adult neurogenesis is present in the adult mammalian CNS but decreases with age and development. In addition, the neurogenic activity in many stem-like cell types decreases with age [46].

After CNS injury, NSPCs become activated and display enhanced proliferation and differentiation potential outside of the normal lineage programming [96]. Ependymal cell activity also varies significantly between major SCI models such as contusion model, hemisection/transection model, and stab model (Figure 4). 

In the stab SCI model, a thin blade penetrates the central canal, disturbing the ependymal cell layer and damaging ependymal cells. This results in high proliferation and contribution to both the glial scar content and lesion by ependymal progeny [51,104]. Images and quantification of this scar border directly within the NSC niche of the central canal may have overinflated the regenerative capacity of ependymal cells in adult mammals. Due to the location of the injury, the segregation of the injured cells and formation of scar tissue consists highly of ependymal cell progeny which can easily integrate into the network of scar tissue in their immediate vicinity. In addition, a greater percentage of damaged ependymal cells in this injury model result in increased proliferation. As a note, the little migration necessary from the NSPC niche to the entire lesion primarily differentiates this injury model from many others and results in high ependymal contribution. 

In the dorsal hemisection or full transection SCI model, a surgical blade is used to slice half, or the entire spinal cord and the central canal ependymal cell populations are damaged. However, these cells still contribute little to the glial scar border due to the limited migratory capacity of ependymal progeny following injury [105]. Resident astrocytes and NG2+ cells in these models contribute to glial scar border, with lesser ependymal progeny recruitment to the injury [106]. The anatomical location of the damaged cells in this model explains the small ependymal contribution to the scar border and neural lesion. Populations primarily line the central canal, and the slice injury minimizes cellular/niche damage. Thus, consistency of damaged or injury-stimulated cell types is little in comparison with stab model (Figure 4).

In the contusion SCI model, the most clinically relevant injury model, a pneumatically driven rod is dropped onto the cord and raised immediately to create the injury. The ependymal cell layer is disturbed, but not penetrated and thus minimal ependymal cell progeny migrate to the site of injury and contribute to the glial scar [107] (Figure 4). While ependymal cells are multipotent and produce oligodendrocyte and astrocyte fated cells, the limited migratory capacity of ependymal cells after injury results in little contribution to the injury itself. The contusion injury does not directly damage ependymal cells but does damage tissue in proximity with the NSPC niche of the central canal. 

Ependymal contribution to glial scar has been associated with age, as younger ependymal cells in the spinal cord retain more proliferative and migratory capacity, thus increased contribution to glial scarring [51]. This is significant in clinically relevant contusion SCI model, where the lesion is not in direct contact with the central canal ependymal layer but may be subject to molecular signals from damaged cells. 

The ependymal cell populations in the brain and spinal cord are highly heterogeneous and can generate neuroblasts and glia in response to stroke, elicit aberrant NSPC activity following SCI and TBI. Within ependymal populations, subpopulations have been identified by gene expression studies [108]. However, the function of these subpopulations is still under investigation, but a thorough understanding will support the development of treatments for stroke, TBI, SCI, schizophrenia, and many other injured or diseased states.

Ependymal cells in the adult mammalian CNS are stem-like cells, as demonstrated by their ability of self-renewal and multipotency (Table 1 and Table 2). They contribute to glial populations in the normal physiology and injury. In the stab SCI model, a blade penetrates the ependymal cell layer in the central canal. In this case, ependymal cells contribute greatly to glial scar border formation and migrate into the neural lesion. However, in all other validated SCI models these cells do not contribute a significant number of astroglial progeny to the glial scar. These cells serve as a major source of NSPCs in the spinal cord but do not provide a suitable therapeutic target in contusion SCI. Regardless, these cells may serve as a viable therapeutic target for regeneration in stab wound type clinical SCI and TBI.

### 4.2. NG2+ Cells

The NG2 is a type I transmembrane glycoprotein also called chondroitin sulfate proteoglycan 4 or nerve glial antigen-2 [124]. The NG2+ cells are heterogeneous populations composed of glia, pericytes and macrophages of vasculature, also regarded as polydendrocytes. Cell morphology varies throughout life and subpopulation, but generally can be characterized by soma with long extended or short processes. A major percentage of the NG2+ glia population are oligodendrocyte progenitor cells (OPCs) which actively contribute to the oligodendrocyte population [116]. Populations of OPCs have been harvested and purified in vitro and approximately 95% are positive for NG2 marker [50]. NG2+ cells have been controversially proposed as stem-like cells in the literature [58]. However, several studies contradict each other regarding this cell type classification and contribution to the pathophysiology of SCI, TBI, and various diseased states. These discrepancies are largely due to differences in recombinant genetic mouse lines, animal models, injury models, and quantification techniques. Transgenic mouse lines have been established to target NG2+ populations using promoters from PDGFRa, Olig2, and Sox10 genes; commonly expressed in NG2+ populations [125]. Approximately 80% or more cells in NG2+ populations are targeted by these factors, but in varying ratios [111]. This fact contributes to inconsistencies between published results on NG2+ populations and NSPC characterization. Targeted populations may not truly represent the NG2+ cells but extend to a variety of other cell types as well [111]. In addition, major differences in injury models and glial scar properties contributes to controversial NG2+ stem-like nature [125]. Issues may also result from astrocyte identification methods within the glial scar border, as GFAP is the most common astrocyte marker used but is not expressed by all astrocyte subtypes.

During development, the NG2 marker exists in three major populations of self-renewing cells: oligodendrocyte lineage, NG2 glia, and astrogenic glia. Within early to mid-stages of development, NG2+ cells have high differentiation potential and supply progeny to populations such as astrocytes and oligodendrocytes [126]. In the adult, there are three well-established glial populations, i.e., astrocytes, microglia, and oligodendrocytes, and NG2+ cells make up the fourth major glial population [50,127,128]. NG2+ polydendrocytes are characterized as highly proliferative [106]. NG2+ cells are evenly distributed throughout brain and spinal cord, often described as a checker pattern in tissue sections of the spinal cord. These cells also interact uniquely with neurons and glia, receiving both inhibitory and excitatory signals from areas throughout the brain indicating the diverse functionality of the NG2+ cell populations [129]. Interestingly, NG2+ cells generate white matter mature oligodendrocyte cells at a faster rate than grey matter mature oligodendrocyte cells. In addition, NG2+ cells in white and grey matter have been shown to exhibit differing morphological and electrophysiological characteristics such as the ability to generate mature action potential spikes in white matter NG2+ cells [120]. 

NG2+ cells isolated from the rat optic nerve exhibited multipotency in vitro as they differentiated into oligodendrocyte or astrocyte cell fates [130]. However, this capacity is limited in vivo, as NG2+ contribute primarily to oligodendrocyte populations [77,123]. Interestingly, ectopic expression of Sox2 has been shown to restore multipotent lineage capacity in the adult [59]. Self-renewal has also been observed in NG2+ cell populations in vivo [131]. Distinct heterogeneous populations of NG2+ cells exist in the adult CNS and may contribute to a complex variety of essential activities, e.g., maintenance of homeostasis, glutamate signaling [124,132]. 

Extrinsic and intrinsic factors guide NG2+ cell activity in the adult, including ciliary neurotrophic factor (CNF), brain derived neurotrophic factor (BDNF), astrocyte derived growth factors, neurotransmitters, cytokines, Notch and Wnt signaling pathways, and transcription factors Olig2 and Sox10 [106,125,133]. The interaction between these signals maintains the delicate balance to maintain the NSPC behavior of NG2+ populations. Quiescence/survival of these populations is regulated by chemokine CXCL12 and CXCR4 [134]. Proliferation is directed by intrinsic gene Ascl1 [135]. Differentiation is directed by Shh signaling in oligodendrocyte populations in the dorsal SVZ, but not the ventral SVZ, indicating regionally distinct NG2+ populations in the SVZ. In addition, oligodendrocyte lineage differentiation is directed by Wnt signaling and the kon-tiki gene. Migration is directed by NG2 interaction, as shown in stab injuries to orient the NG2+ cells toward the injury site [136]. Integration is directed by neurotransmitter receptor activation and synaptic input from glutamatergic and GABAergic neurons. 

NG2+ cell populations react to traumatic injury and neurodegenerative/demyelinating conditions. Specifically, traumatic injury reactivates the differentiation potential in populations of the brain and spinal cord. NG2+ cells transition to a stem-like state and are assumed to contribute to the injury in two segregated zones, i.e., necrotic core and glial scar. The ratio of infiltrating cells and percentage NG2+ contribution to total scar border varies significantly between injury types. For example, in the contusion SCI model, 25% NG2+ progeny cells were reported in the glial scar border [57]. This may be due to increased inflammation and extent of injury resulting in greater differentiation potential. In the dorsal hemisection model, 5% NG2+ astrocyte progeny content has been reported in the glial scar border [51]. In the cortical stab model, 8% NG2+ cell progeny content has been reported in the glial scar border [58]. Thus, the highest NG2+ contribution to the glial scar border occurs in the contusion injury model (Figure 4). This may be due to the size and depth of lesion formed in clinically relevant contusion SCI.

In acute injury, NG2+ cells migrate into the injury site and contribute to the formation of the glial scar border. Specifically, the migratory capacity of NG2+ cells increase with traumatic injury in both the brain and spinal cord [137,138]. This may be due to signals directing aberrant migratory activity or regenerative abilities of NG2+ cells in the spinal cord. In the formation of the glial scar border, Wnt signaling controls multipotent differentiation into astrocyte and oligodendrocyte cell types [139]. In addition, NG2+ progeny cells in the spinal cord preferentially differentiate into reactive astrocytes via Shh signaling.

NG2+ cells are a potential target for SCI, TBI, MS, and other demyelinating and degenerative conditions due to the consistency of NG2+ cells throughout the neurogenic niches of the CNS. NSPC genes such as Sox2, Olig2, Pax6, and PDGFRa have been forcefully overexpressed in NG2+ populations and resulted in increases in neurogenesis and reactivation of stem-like characteristics [126]. The clinical relevance of this approach has recently been demonstrated, i.e., targeted overexpression of Sox2 in NG2+ populations resulted in improved functional recovery after SCI [59]. Neurogenesis in the post injury CNS is altered significantly leading to ineffective maintenance of homeostasis, learning, memory, and generation of nonfunctional neuronal networks.

Overall, although the NG2+ cells are not inherently stem-like cells, they are self-renewing during development and into adulthood (Table 1 and Table 2). In addition, NG2+ cells are widely known to actively proliferate and contribute to oligodendrocyte glial cells in the mammalian CNS. NG2+ cells are not multipotent in nature, however, the multilineage potential is reactivated by injury and mediators such as transcription factors. This capability of NG2 cells to acquire multipotency has great potential to contribute to neural regeneration. Thus, these cells present a viable therapeutic route to modulate the glial scar formation in SCI and TBI. Gene or cell therapy targeted for NG2+ cells may use the acquired self-renewal capability of these populations to regenerate tissue in the CNS. The broad distribution of these cells throughout the CNS may increase the application and ability of targeted stem cell activity to treat injured and diseased states. Other feasible therapeutic approaches include applying neurotrophic, growth, and anti-inflammatory factors to guide the activity of NG2+ cells after injury.

## 5. Conclusions

A wide variety of NSPCs reside within the adult CNS. This diversity contributes to the complex pathophysiology of clinical injured and diseased states of the CNS such as SCI, TBI, and retinal degeneration. Adult neurogenesis in the brain, spinal cord, and retina is necessary for maintenance of homeostasis, learning, memory, and energy metabolism. Interestingly, variability exists between major neurogenic niches of the CNS in the retina, brain, and spinal cord. Differences in specification of these cells, ratio of cells, and existence at all or some of these cell types in the neurogenic individual niches varies greatly with anatomical location [28].

NG2+ and ependymal cells are heterogeneous populations distributed distinctively in the CNS neural niches. During development, NG2+ cells are multipotent and self-renewing, contributing to glial cell populations including astrocytes and oligodendrocytes. Multipotency is lost in postnatal stages but is acquired following traumatic injury and by manipulation of gene expression, thus these cells present a highly viable target for traumatic CNS injury. The relative percentage of NG2+ cells contributing to the glial scar is highest in the contusion SCI model, lesser in the cortical stab SCI model, and the least in the hemisection SCI model. During development, ependymal cells proliferate and contribute to multipotent cell types. In adults, ependymal cells maintain multipotency and activation, but to a lesser degree than in embryonic development. After contusion and hemisection injury, ependymal cells contribute minimally to glial scar formation due to lack of direct damage to the central canal of the spinal cord. However, stab injury damages only the central canal, leading to scar border formation directly inside of the ependymal cell layer. Injured ependymal cells proliferate actively and produce multipotent progeny to contribute to glial scar border and the neural lesion extensively. However, migratory capacity is seemingly unaffected by injury. Thus, these cells present a therapeutic target for specific injury types which exclusively damage ependymal cell layer. Current and previous NSPC-based clinical trials for SCI, TBI, mechanical/chemical injury, and diseased states such as Glaucoma, Retinitis Pigmentosa, demyelinating diseases, and inflammatory conditions are summarized in Table 3. As another therapeutic route, the migratory capacity of ependymal progeny can be stimulated to contribute to the site of injury. Examples include stimulation of migration via neurotrophic/growth factors and gene/cell therapy. 

Interestingly, ependymal cells exhibit inverse behavior to NG2+ cells after traumatic injury (Figure 4), contributing most to the glial scar in stab SCI model and the least in contusion SCI model. This may be due to the anatomical location of these cell populations and limited migratory capacity of ependymal progeny. Thus, targeting ependymal and NG2+ cells for CNS regeneration, as well as a variety of other proliferating cell types such as sub-populations of astrocytes, represent potential therapeutic strategies for regenerative medicine. However, this will depend on the extent, anatomical location, and type of injury or disease.

Future research directions for NG2+ and ependymal cell populations include a deeper mechanistic understanding of progeny differentiation fates, migratory ability, and functionality of subpopulations of these cells in the normal physiology and traumatic injury. Applicable fields include developmental biology, tissue engineering and regenerative medicine. Application of these cell types as target cells may hold the key to treatments for SCI, TBI, retinal mechanical damage, and degenerative diseases of the CNS. The data/information presented in this review is derived from rodent models. Data from other species are specified in the text.

## Figures and Tables

**Figure 1 cells-10-02045-f001:**
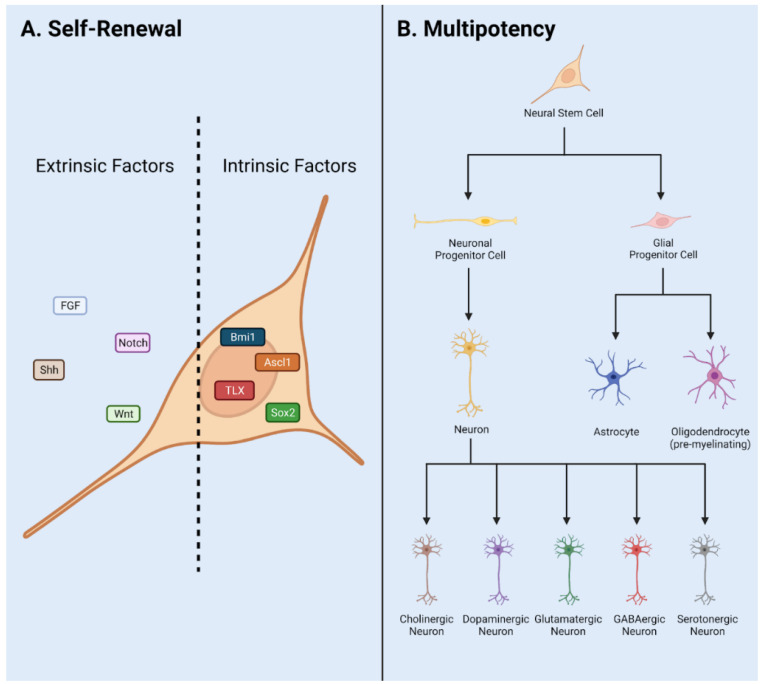
NSPC characteristics in adult mammals. (**A**) Self renewal requires input via extrinsic and intrinsic factors. These include signaling pathways Notch, Wnt, and Shh, and transcription factors Sox2, Ascl1, Bmi1, Tlx, and neurotransmitters and neurotrophic/trophic growth factors. (**B**) Multipotency allows NSPCs to differentiate into a variety of cell fates such as Neurons, Astrocytes, and Oligodendrocytes. Adapted from Navarro Quiroz et al., 2018 [6].

**Figure 2 cells-10-02045-f002:**
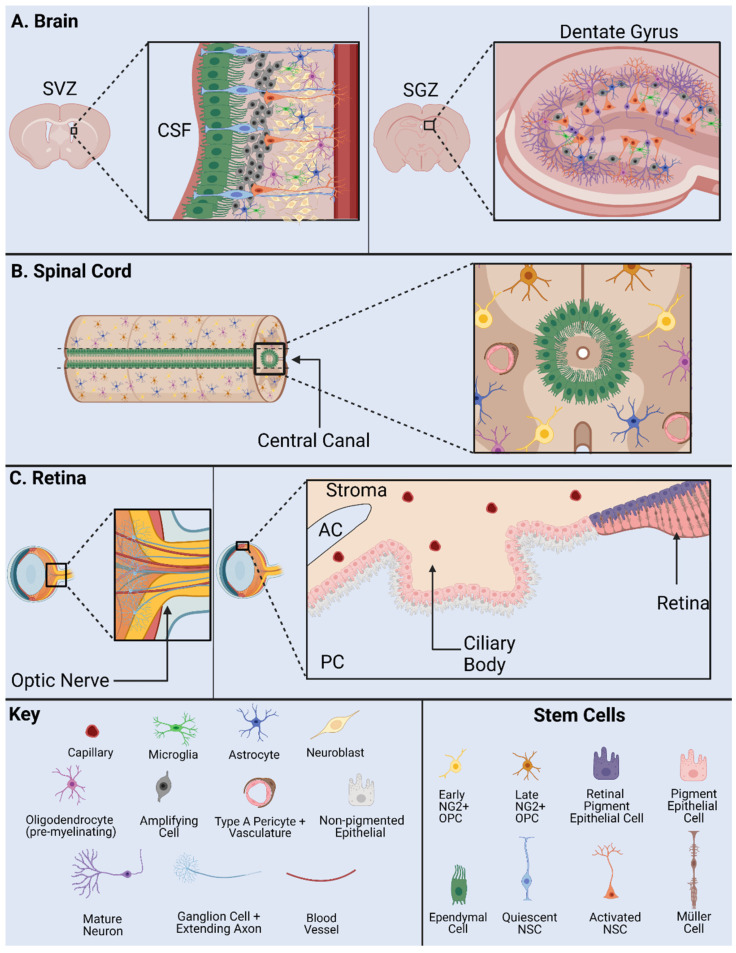
NSPC Niche in mammals: the SVZ and SGZ in the brain (**A**); the ependymal cells and NG2 cells in the spinal cord (**B**); and the base of the optic nerve, the Müller glia, and the pigment epithelium in the retina (**C**). AC, anterior chamber; CSF, cerebrospinal fluid; PC, posterior chamber; SVZ, subventricular zone; SGZ, subgranular zone. Adapted from Cutler and Kokovay, 2020 [26] (**A**); Sabelström et al., 2014 [27], Andreotti et al., 2019; Picoli et al., 2019 [28,29] (**B**); Yoshida et al., 2000 [30] (**C**).

**Figure 3 cells-10-02045-f003:**
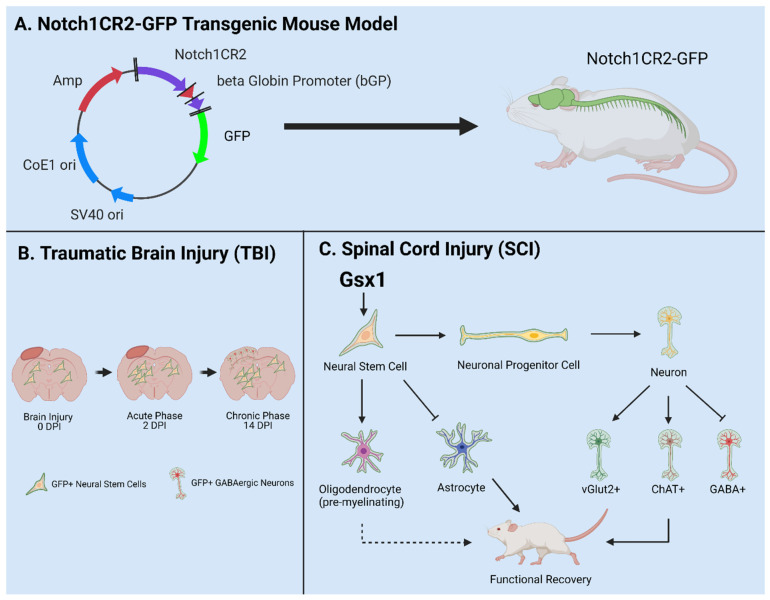
Utilities of the Notch1CR2-GFP transgenic mouse line in SCI and TBI models. (**A**) Notch1CR2-GFP transgenic mouse model labels NSPCs in the CNS. (**B**) Adult NSPCs in the brain proliferate in the acute phase of TBI and differentiate into neurons in the chronic phase of TBI. (**C**) In the injured spinal cord, Gsx1 expression promotes adult NSPC proliferation and preferential differentiation into excitatory interneurons and inhibits astrocytes and glial scar formation after injury. Adapted from Tzatzalos, et al., 2012 [81] (**A**), Anderson et al., 2020 [80] (**B**) and Patel et al., 2021 [79] (**C**).

**Figure 4 cells-10-02045-f004:**
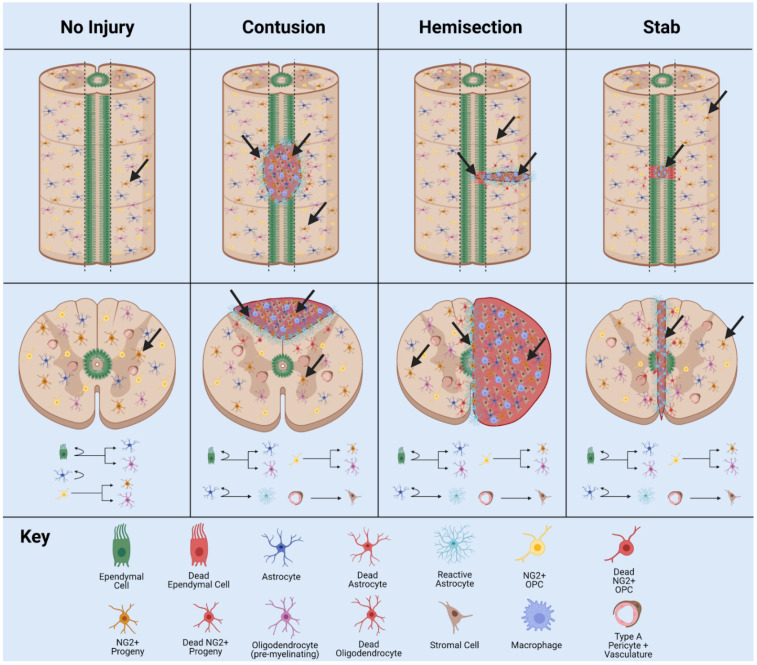
Behavior of ependymal cells and NG2+ cells in animal models of SCI. In normal physiology, the ependymal cells lining the wall of the central canal are largely quiescent, while NG2+ cells are ubiquitously distributed throughout the grey and white matter of the spinal cord. In contusion SCI, the ependymal cell layer is not damaged, but may increase proliferation and differentiation potential. In the hemisection model, the ependymal cell layer is damaged and ependymal cells/NG2+ cells are activated by injury. In stab SCI, the ependymal cell layer is damaged and contributes greatly to glial scar formation. Adapted from Sabelström et al., 2014 [27], Hackett et al., 2016 [103], and Picoli et al., 2019 [29].

**Table 1 cells-10-02045-t001:** Ependymal and NG2+ cell activity: normal physiology vs. injury.

Ependymal	No Injury	Contusion	Hemisection	Stab
Proliferation	Yes, Medium [52] Yes [51]	Yes, High [109]	Yes [95] Yes, Medium [51]	No: Ependyma not injured [91] Yes, Low: Ependyma injured [91] Yes [110]
Differentiation	No [51]	Yes [109]	Yes [95] Yes [51]	Yes, Low [91] Yes [110]
Migration	Yes, Low [52]	Yes, Low [109]	Yes, High [95]	Yes, Low [91] Yes [110]
Quiescence	Yes [111] Yes [51]	No	No	No
Glial Scar Formation	N/A	Yes [109]	No [51]	Yes: Ependyma injured [91]
Neural Lesion	N/A	Yes [109]	Yes [51]	Yes [91]
**NG2**	**No Injury**	**Contusion**	**Hemisection**	**Stab**
Proliferation	Yes, Gradual Decline [112] Yes, High [111]	Yes, Low [109] Yes, High [57] Yes, High [113]	Yes, High [57] Yes, High [114]	Yes, High [115]
Differentiation	Yes, Medium [112]	Yes, Medium [106]	Yes, Medium [116] Yes, Medium [57]	Yes, Low [106] Yes [115]
Migration	Yes, Medium [112]	Yes [113]	Yes [57]	Yes [116] Yes [115]
Quiescence	Yes, Low [116] Possibly [21]	No [113]	No [114]	Decrease [117]
Glial Scar Formation	N/A	Yes [57] Yes, 25% [58]	Yes [116] Yes, 5% [58]	Yes, 5–8% [58] Yes [115]
Neural Lesion	N/A	Yes, High [113]	Yes, Delayed increase [118] Yes [114]	Yes [118] Yes [115]

Ependymal and NG2+ cell stem-like behaviors in the normal physiology and after different types of SCI.

**Table 2 cells-10-02045-t002:** Literature supporting or refuting NG2+ and ependymal cells as stem cells.

Ependymal	For	Against
Capable of Division or Self-Renewal	[95,119,120,121]	[97,122]
Capable of Giving Rise to Specialized Cells	[95,108,119,121]	[91,97]
Expression of Stem Cell Markers	[97,108,119]	[59]
**NG2**	**For**	**Against**
Caple of Division or Self-Renewal	[106,111,123]	[50]
Capable of Giving Rise to Specialized Cells	[59,106,111,123]	N/A
Expression of Stem Cell Markers	[57,59]	[50]

References in support and against NG2+ cells and Ependymal cells as stem cells in the CNS.

**Table 3 cells-10-02045-t003:** NSPC-based clinical trials.

Title	Start–End	Conditions	Intervention
Pilot Investigation of Stem Cells in Stroke	Jun. 2010–Mar. 2023	Stroke	Biological: CTX0E03 neural stem cells
Study of Human Central Nervous System Stem Cells (HuCNS-SC) in Patients With Thoracic Spinal Cord Injury	Mar. 2011–Apr. 2015	Thoracic SCI	Biological: HuCNS-SC cells
Human Neural Stem Cell Transplantation in Amyotrophic Lateral Sclerosis (ALS) (hNSCALS)	Dec. 2011–Dec. 2015	Amyotrophic Lateral Sclerosis	Biological: Human Neural Stem Cells
Study of Human Central Nervous System Stem Cells (HuCNS-SC) in Age-Related Macular Degeneration (AMD)	Jun. 2012–Jun. 2015	Macular Degeneration	Drug: HuCNS-SC cells
Intrathecal Administration of Autologous Mesenchymal Stem Cell-derived Neural Progenitors (MSC-NP) in Patients With Multiple Sclerosis	Apr. 2014–Mar. 2017	Multiple Sclerosis	Biological: intrathecal administration of autologous MSC-NP
Pilot Investigation of Stem Cells in Stroke Phase II Efficacy (PISCES-II)	Jun. 2014–16 Aug. 2017	Ischaemic Stroke/Cerebral Infarction/Hemiparesis/Arm Paralysis	Biological: CTX DP
Safety Study of Human Spinal Cord-derived Neural Stem Cell Transplantation for the Treatment of Chronic SCI (SCI)	Aug. 2014–Dec. 2022	SCI	Drug: Human spinal cord stem cells.
NeuroRegen Scaffold™ Combined With Stem Cells for Chronic Spinal Cord Injury Repair	Jan. 2016–Dec. 2021	SCI	Biological: NeuroRegen scaffold/neural stem cells transplantation
Safety Study of Human Neural Stem Cells Injections for Secondary Progressive Multiple Sclerosis Patients (NSC-SPMS)	9 Sept. 2017–29 May 2021	Multiple Sclerosis	Biological: Human Neural Stem Cells
Intrathecal Administration of Autologous Mesenchymal Stem Cell-derived Neural Progenitors (MSC-NP) in Progressive Multiple Sclerosis	21 Sept. 2018 –Nov. 2023	Multiple Sclerosis	Biological: Intrathecal MSC-NP injection/Other: Intrathecal saline injection
Use of Mesenchymal Stem Cells (MSCs) Differentiated Into Neural Stem Cells (NSCs) in People With Parkinson’s (PD).	Jun. 2018–Sept. 2020	Parkinson Disease	Biological: Injection of Umbilical cord derived MSCs
CNS10-NPC for the Treatment of RP	Mar. 2020–Jun. 2022	Retinitis Pigmentosa	Biological: CNS10-NPC implantation
A Safety and Tolerability Study of Neural Stem Cells (NR1) in Subjects With Chronic Ischemic Subcortical Stroke (ISS)	4 Jan. 2021–31 Dec. 2024	Ischemic Stroke	Biological: Neural Stem Cells
Transplantation of Neural Stem Cell-Derived Neurons for Parkinson’s Disease	Jun. 2021–30 Jun. 2023	Parkinson’s Disease	Biological: Intracerebral microinjections

Recruiting, current, and past NSPC-based clinical trials (retrieved from ClinicalTrials.gov, accessed on 14 July 2021).

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
