# Peer review of "Diversity of Adult Neural Stem and Progenitor Cells in Physiology and Disease"

_cells, 2021, doi:10.3390/cells10082045_

Round 1

Reviewer 1 Report

In this review, the authors attempted to discuss the heterogeneous neural stem and progenitor cell (NSPC) populations in the normal physiology and to highlight their potentials and roles in injured/diseased states for regenerative medicine. Although this paper deals with an interesting topic, consistency between the first and the second half is insufficient. My comments are below.

Major points

The first and second paragraphs evenly include information regarding NSPCs in the brain, spinal cord, and retina. These paragraphs well reflect the present title. On the other hand, the fourth paragraph almost consists of information regarding ependymal and NG2+ cells in spinal cord injury. Understanding of NSPCs is important for not only spinal cord injury but also neuronal and retinal degenerative diseases including Parkinson’s disease and age-related macular degeneration. One suggestion is that information regarding NSPCs in neuronal and retinal degenerative diseases is increased and that information regarding ependymal and NG2+ cells in spinal cord injury is decreased. Another suggestion is that the title is changed to ‘Ependymal and NG2+ cells in physiology and spinal cord injury’ and that information regarding ependymal and NG2+ cells in the spinal cord are highlighted in the first and second paragraphs.

Minor points

  1. Astrocytes have competency of neural stem cells in the SVZ of adult?

  1. In the Figure 2A, many types of cells are shown in the SVZ, compared to SGZ. Astrocytes and microglia do not regulate function of NSPCs in the SGZ?

  1. Vein and artery shown in the Key cannot be found in the Figure 2A-C.

  1. In the Figure 3B, GFP +Neural Stem cells is probably ‘GFP+ Neural Stem cells’.

Reviewer 2 Report

The present review article entitled “Diversity of adult neural stem and progenitor cells in physiology and disease provided summary of the information relevant to different types of NSPC and their potential application as potential cellular  therapy for different neurodegenerative and traumatic insults of CNS.

The is a very interesting topic specially its potential application on regenerative neurology.

Despite interesting topic/aims, there are several points that are needed to be addressed before final acceptance of the manuscript:

  1. The authors should provide a list of current clinical trials using different types of NSPC for cell-based therapy for different CNS and retinal insults/diseases.
  2. A section about the safety and of use of different types of NSPC should be provided.
  3. A section about potential genetic manipulation of NSPC as a mean to enhance their proliferation and differentiation potential would be crucial for this review.

Reviewer 3 Report

The MS entitled “Diversity of adult neural stem and progenitor cells in physiology and disease” aims to review a subject of interest in neuroscience and medicine. In this review, the diversity of the adult neural stem and progenitor cells is presented in four regions of the mammalian central nervous system: cerebrum (SVZ and SGZ), retina and spinal cord. It is not clear to what species the description refers, as data from mouse and human are mixed and no any clarification is provided.

The biggest problem in this review is the confusion of what neurogenesis means. Neurogenesis is the formation of neurons from neural stem cells occurring during embryonic development and throughout adult life. Indeed, in adult life, neurogenesis is restricted to specific neurogenic niches and can be stimulated or extended in different conditions, which is partially addressed in this review.

The review starts with the definition of the neural stem and progenitor cells during development: “During development, neural stem cells (NSCs) are responsible for the formation of the central and peripheral nervous systems. Initially NSCs, also called radial glial cells, proliferate into pools of neural progenitor cells (NPCs)”.  Several mistakes are present in this statement. Neural stem cells (NSCs) are not responsible for the formation of the peripheral nervous system, but other early derivatives of the ectoderm, the neural crest stem cells and the preplacode progenitors; the initial NSCs are called neuroepithelial cells, which differentiate into radial glial cells; radial glial cells can generate different types of transit amplifying progenitors.  “These progenitors then migrate and differentiate into highly specified networks of neurons, oligodendrocytes, and astrocytes via neurogenesis [1].” Here the main confusion, as already specified, regards the neurogenesis. The oligodendrocytes and the astrocytes are formed via gliogenesis.

While the NSCs, which can produce neurons and glial cells, are restricted to few zones in the adult CNS, the neural progenitors forming glial cells are presented in all areas of the adult CNS.

This review needs extensive reformulation in order to correctly address the human neural stem and progenitor populations and their fates in physiology and disease.

Round 2

Reviewer 1 Report

The authors have almost appropriately revised the manuscript according to the comments made by the reviewer. I still hold mild reservations on the acceptability of the manuscript in its present form. Before publication, I would like the authors to address the following points.

  1. The citation of the Tables should be shown in the text.

  1. Abbreviations such as TBI, SCI, and MS, should be spelled out at the first appearance.

Reviewer 2 Report

The authors have added the requested information